# FLuID: Mitigating Stragglers in Federated Learning using Invariant Dropout

**Irene Wang**[1,3], **Prashant J. Nair**[1], **Divya Mahajan**[2,3]
University of British Columbia[1], Microsoft[2], Georgia Institute of Technology[3]
irene.wang@gatech.edu   prashantnair@ece.ubc.edu   divya.mahajan@gatech.edu

## Abstract

Federated Learning (FL) allows machine learning models to train locally on individual mobile devices, synchronizing model updates via a shared server. This approach safeguards user privacy; however, it generates a heterogeneous training environment due to the varying performance capabilities of devices. As a result, "straggler" devices with lower performance often dictate the overall training time. In this work, we aim to alleviate this performance bottleneck due to stragglers by dynamically load balancing the training across the system. We introduce *Invariant Dropout*, a method that extracts a sub-model based on the weight update threshold, thereby minimizing potential impacts on accuracy. Building on this dropout technique, we develop an adaptive training framework, Federated Learning using Invariant Dropout (FLuID). FLuID offers a lightweight framework for sub-model extraction to regulate the computational intensity, thereby reducing the load on straggler devices without affecting model quality. Our method leverages neuron updates from non-straggler devices to construct a tailored sub-model for each straggler based on client performance profiling. Unlike prior work, FLuID can dynamically adapt to changes in stragglers as runtime conditions shift. We evaluate FLuID using five real-world mobile clients. The evaluations show that Invariant Dropout maintains baseline model efficiency while alleviating the performance bottleneck of stragglers through a dynamic and lightweight runtime approach. [1]

## 1 Introduction

Federated Learning (FL) enables machine learning models to train on edge and mobile devices, synchronizing the model updates via a common server [SB09, SVHN10, LSTS20, ZBW+22]. This method ensures user privacy as local data remains on the device, minimizing the risk of data breaches, leaks, or unauthorized access to sensitive information in the cloud [MMR+17]. However, FL introduces heterogeneity due to varying performance capabilities of the participating devices. Consequently, straggler devices with lower computational and network performance often dictate the overall training latency and throughput, as shown in Figure 1.

Previous research mitigates the impact of stragglers through asynchronous aggregation, where clients can communicate and update the server model independently and asynchronously [CCA+21, CNSR20, XKG19]. While this approach alleviates some of the detrimental effects of stragglers, it can introduce staleness into the global model. This occurs because the gradients used to update the server model may rely on outdated parameters, resulting in inaccuracies that have the potential to slow down convergence and reduce overall accuracy [BHS20, CCA+21, WUH+21, HBGS19].

Other research proposes eliminating updates from slower devices entirely. [KMA+19]. However, this approach can introduce training bias, as it effectively excludes certain clients and their respective data.

---

[1]Source code is available at https://github.com/iwang05/FLuID

37th Conference on Neural Information Processing Systems (NeurIPS 2023).

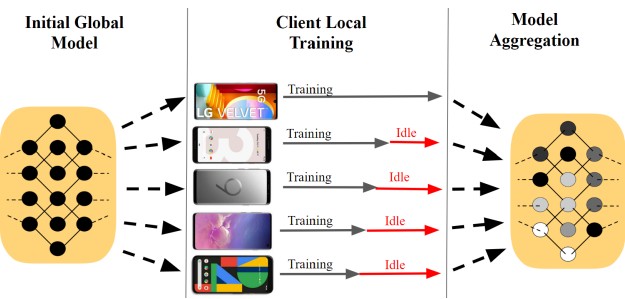

Figure 1: Straggler's impact on FL performance. In synchronous FL, all clients, including stragglers, participate in global model aggregation.

To ensure the contribution of all devices, recent works in this area employ a dropout technique where the stragglers only train a subset of the global model [HLA+21, MGZP22]. The overall accuracy of the model is determined by the subset of neurons that are dropped from the global model. Thus, previous work alleviates the training load on stragglers by either incurring training bias, creating performance-centric sub-models, or entirely reconstructing the sub-model.

Our paper proposes a novel dropout technique called Invariant Dropout, which identifies "invariant" neurons—those that train quickly and show little variation during training. We observe that the training (computational load) and transfer (communication load) of these invariant neurons to and from the straggler devices contribute minimally to the efficiency of the global model. Thus, they can be dropped. We observe that after only 30% of the training iterations, 15%-30% of the neurons become invariant across CIFAR10 [Kri09], and LEAF datasets FEMNIST, and Shakespeare [CDW+18]. Building on this insight, we develop a dynamic framework called Federated Learning using Invariant Dropout (FLuID), which adjusts the sub-model size based on both the magnitude of neuron updates and the computational capabilities of the client devices. In addition to introducing a new dropout technique, FLuID, unlike prior work, periodically calibrates the sub-model size during runtime to account for changes in stragglers due to factors like low battery or network issues.

Overall, we face two challenges in building FLuID — identifying invariant neurons and establishing a dynamic lightweight method for straggler identification and sub-model size determination at runtime. FLuID addresses the first challenge by leveraging the non-straggler clients, which typically outnumber stragglers and train on the entire model, to identify invariant neurons. The server is not used for this purpose as it receives updates from stragglers running a sub-model. For the second challenge, FLuID uses a drop-threshold, below which neurons are dropped, allowing dynamic sub-model determination for each straggler. By profiling client training times and identifying stragglers during the initial epochs of each calibration step, FLuID can incrementally adjust the drop-threshold until the number of selected neurons matches the target sub-model size for stragglers.

Our evaluation of FLuID on various models, datasets, and real-world mobile devices shows an up to 18% speedup in performance. Additionally, it improves training accuracy by a maximum of 1.4 percentage points over the state-of-the-art Ordered Dropout, all while mitigating the computational performance overheads caused by straggler devices.

## 2    Related Work

In the domain of heterogeneous client optimization, several prior work have tried to mitigate the effects of stragglers.

**Dropout techniques.** Federated Dropout [CKMT18, RKPH22] randomly drops portions of the global model, sending a sub-model to the slower devices. However, this can potentially impact accuracy. Subsequent work, Ordered Dropout [HLA+21, DDT21], mitigates this accuracy loss by regulating which neurons are dropped, either from the left or right portions of the global model. Other methods, such as those outlined in [MGZP22], use a low-rank approximation to identify over-parameterized neurons and generate tailored sub-models. Despite these advancements, none of these works consider the individual contribution of each neuron while creating a sub-model. In contrast, our work not only introduces a novel dropout technique that takes into account the contribution of each neuron but also

provides a framework capable of dynamically identifying slower devices and adjusting the dropout rate accordingly.

**Server offloading strategies using split learning.** The approach proposed by [WUH+21] employs split learning to offload part of the model to the server, while [HAA20] transfers their knowledge to a larger server-side CNN. [UWH+21] focuses on device mobility during Federated Learning, offloading training to edge servers. Finally, [WQR+22] introduces a compression method for split learning. However, unlike these methods, Invariant Dropout does not require data transfer out of the device and into the server, instead conducts all training on client devices. Nonetheless, if data movement was a possibility, Invariant Dropout can be stacked on top of these techniques to determine which part of the model needs to be offloaded to the server.

**Communication Optimizations.** To reduce learning time, [HWL20] proposes an online learning approach that reduces the communication overheads of Federated Learning by adaptively sparsifying gradients. [CSP+21] introduces a Federated Learning framework utilizing a probabilistic device selection method to improve FL convergence time. [XFD+21] proposes a framework incorporating overhead reduction techniques for efficient training on resource-limited edge devices, including pruning and quantization. Unlike these works, Invariant Dropout avoids both communication and computation overhead by dropping the least contributing "invariant" neurons, presenting a new insight compared to these prior works.

**Model Pruning.** PruneFL [JWK+22] introduces an approach to dynamically select model sizes during FL and reduce communication and computation overhead, thereby minimizing training time. Work in [MSA+21] prunes the global model for slower clients by excluding neurons with no or small activations. These approaches either generate a single sub-model for all clients, including non-stragglers, to train on or generate a static sub-model for the entire training process which can result in the stragglers permanently losing the opportunity to contribute to certain parts of the model. In contrast, Invariant Dropout enables the generation of multiple sub-models with various resource budgets and dynamically chooses a sub-model for each round based on current neuron contribution.

**Coded Federated Learning.** CodedPaddedFL and CodedSecAgg [SKRA21] utilize coding strategies to mitigate the impact of slower devices. By sharing an encoded version of their data with other clients, both schemes introduce redundancy in the client's local data. Therefore, during training, the computation of a subset of clients is sufficient to train the global model and the computations of straggling clients can be discarded without loss of information. While CodedPaddedFL combines one-time padding with gradient codes, codedSecAgg is based on Shamir's secret sharing. Unlike these methods, Invariant Dropout allows each client to keep their data local.

**Training a Global Family of Models.** SuperFed [KALT23] proposes co-training of multiple models to reduce training costs. It achieves this by sending subnetworks of different sizes to all clients. In contrast, FLuID focuses on optimizing the performance of stragglers by mostly training clients on the full global model and a smaller percentage (stragglers) on sub-models tailored to their capabilities.

## 3 Background and Motivation

### 3.1 Challenges in Federated Learning

In Federated learning's synchronous aggregation protocols, the server hosts a global model and regularly aggregates updates from clients, which are then redistributed to clients [CCA+21]. Thus, system heterogeneity remains a significant challenge in FL [DTS+22, HLA+21, LSTS20]. As shown in Figure 2a, we observe significant differences in training times across five Android-based mobile phones, from 2018 to 2020, engaged in Federated Learning without any dropout mechanism in place. Depending on the dataset and machine learning model, training time can vary significantly, with differences of up to twofold across datasets such as CIFAR10, FEMNIST, and Shakespeare. The standard deviation between the training times of each client is 0.5, 22, and 21 seconds for FEMNIST, CIFAR10, and Shakespeare, respectively. This highlights the real-world challenge of system heterogeneity, where different devices, even if they are of a similar class (e.g., Android-based mobile phones) with just a few years' difference, can offer dramatically different computational performance. Furthermore, we find the performance of mobile devices can fluctuate over time due to varying network bandwidth and resource availability. In response to these observations, we develop

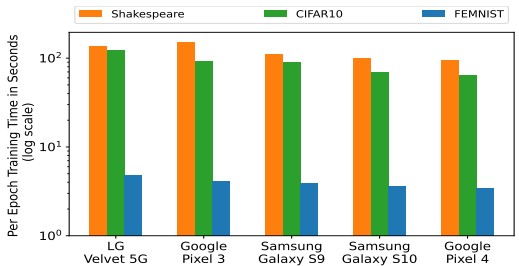
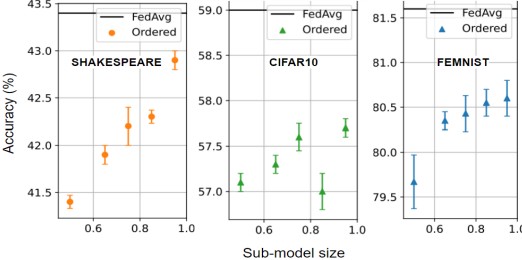

(a) The per-epoch training time of client devices in log scale.

(b) Compared accuracy of Ordered Dropout with baseline implementation of Federated Learning.

Figure 2: (a) Performance variation across mobile devices (in log scale), (b) accuracy implications of prior dropout (static) techniques.

FLuID, a system designed to dynamically identify stragglers and adjust the training load to balance performance with non-straggler devices.

## 3.2 Dropout Techniques

Model dropout is a technique that involves sending a subset of the global model, known as a sub-model, to stragglers for load balancing. There are two state-of-the-art proposals in this space: Federated Dropout [CKMT18] and Ordered Dropout from FjORD [HLA+21]. Federated Dropout randomly drops neurons, simplifying the selection of sub-models, but reducing the global model's accuracy. In contrast, Ordered Dropout systematically drops neurons by maintaining order within the sub-model. These works demonstrate that the sub-model's neuron selection is crucial for the global model's accuracy. In the context of this work, "neurons" refer to filters in convolutional (CONV) layers, activations in fully-connected (FC) layers, and hidden units in Long Short-Term Memory (LSTM) [HS97] layers.

**Accuracy Implications with dropout:** Figure 2b compares the testing accuracy of a non-dropout vanilla Federated Learning (FL) implementation with Ordered Dropout. This comparison is performed using five mobile devices, one of which is a straggler, across three different datasets: CIFAR10 [Kri09], FEMNIST, and Shakespeare [CDW+18]. As the sub-model size decreases, we observe that across all three datasets and machine learning models, Ordered Dropout experiences up to a 2.5 percentage point drop in accuracy. We vary the sub-model size from 0.5 (representing 50% of the global model) to 1 (the entire model). The results with 50-100 clients and 20% of them being stragglers are provided in Section 6.1.

## 4 Invariant Dropout

Invariant Dropout enables stragglers to only train on a sub-model consisting of neurons that 'vary' over time and contribute to the global model. Invariant dropout achieves this by selecting a subset ($i$) of sub-models ($s_i$) from a total sub-model distribution ($S$), where each sub-model represents a different number of neurons and thus varies in compute and memory requirements. Appendix A.1 illustrates the temporal variation in the percentage of invariant neurons in the evaluated models. Note that all clients including the straggler still perform inference on the full model.

## 4.1 Proposed dropout mechanism

Invariant Dropout selects a sub-model from a set of sub-models, $S$, composed of $m$ total sub-models, denoted as $S = s_1, s_2, ..., s_m$. Each sub-model $s_i$ contains neuron layers $a_1, a_2, ..., a_k$. The size of the sub-model corresponds to a dropout rate $r$ that is based on the drop-threshold $th$ for the dropout mechanism. Given a change in neuron value, $g = \Delta a$, we can select the sub-model $s_i = \{a_1, a_2, ..., a_k\}$ where the updates in neurons within the sub-model satisfy $g \geq th \, \forall \, a_j \in s_i$ and $1 \leq j \leq k$. FLuID, which we describe below, selects $s_i$ such that the compute utilization and memory footprint can effectively mitigate the inefficiency of the straggler.

Next, we assume that the system includes $C$ clients with $T$ stragglers and $N$ non-stragglers. $T$ and $N$ are exclusive and independent subsets of $C$, where $T \cup N = C$ and $T \cap N = \phi$. Invariant Dropout maintains a dropout rate $r \in (0, 1]$ per layer across the $T$ stragglers. However, the complexity of selecting $a_j(t + 1)$ will vary with each sub-model $s_i$. To reduce the computational overhead of selecting the appropriate sub-model, ID leverages non-straggler clients to provide directions on the set of $a_j(t + 1)$. The server computes over a subset of potential sub-models $s_i$ to select the one with the maximum updates to the neurons.

## 4.2 Variance in Gradients with Invariant Dropout

Consider $C$ clients participating in Federated Learning. The training data is represented as $x_c c = 1^C$, where $c$ denotes one client, and each client has a corresponding loss function $f c_{c=1}^C$. Learning minimizes the loss function using the following optimization: $f(w) := \frac{1}{C} \sum c = 1^C f_c(w)$, where $w_{t+1} = w_t - \eta_t(g(w_t))$.

Invariant Dropout can be viewed as a sparse stochastic gradient vector, where each gradient has a certain probability of being dropped. The gradient vector is denoted as $G = [g_1, g_2, ..., g_k]$ where $g \in R$ and each gradient has a probability of being retained and transmitted across the network, represented as $[p_1, p_2, ..., p_k]$. The sparse vector for the stragglers is represented as $G_s$. The variance of the ID-based gradient vector can be represented as $E(G_s^2) = \sum_{i=1}^k (g_i^2 p_i)$ [WWLZ18, XYXC21]. The variance of the dropout vector is a small factor deviation from the non-dropout gradient vector represented as follows:

$$\min \sum_{i=1}^k p_i : \sum_{i=1}^k (g_i^2 p_i) = (1 + \epsilon) \sum_{i=1}^k g_i^2 \tag{1}$$

Invariant Dropout drops weights based on the $g = \Delta a$ gradient across epochs. Thus, the probability of a gradient not getting dropped ($p_i$) is inversely proportional to the dropout rate $r$. This implies if $|g_i| > |g_j|$ then $p_i > p_j$. Let's assume that the top-k magnitude of gradients are not dropped. Hence if $G = [g_1, g_2, ..., g_m]$ is sorted, then $G = [g_1, g_2, ..., g_k]$ has a $p = 1$, whereas $G = [g_{k+1}, g_{k+2}, ..., g_d]$ has a probability of $p_i = \frac{|g_i|}{r}$. This modifies the optimization problem in Equation 1 to:

$$\sum_{i=1}^k g_i^2 + \sum_{i=k+1}^m \frac{|g_i|}{r} - (1 + \epsilon) \sum_{i=1}^m g_i^2 = 0, \frac{|g_i|}{r} \le 1 \tag{2}$$

Which implies that

$$r = \frac{\sum_{i=k+1}^m |g_i|}{(1 + \epsilon) \sum_{i=1}^m g_i^2 - \sum_{i=1}^k g_i^2} \tag{3}$$

As per the constraint $\frac{|g_i|}{r} \le 1$:

$$|g_i|(\sum_{i=k+1}^m |g_i|) \le (1 + \epsilon) \sum_{i=1}^m g_i^2 - \sum_{i=1}^k g_i^2 \tag{4}$$

Invariant Dropout retains the gradients with the greatest magnitude. As a result, the boundedness of the expected value from Equation 1 can be represented as follows:

$$\sum_{i=1}^m p_i = \sum_{i=1}^k p_i + \sum_{i=k+1}^m p_i \tag{5}$$

$$\sum_{i=1}^m p_i = k + |g_i| \sum_{i=k+1}^m \left( \frac{\sum_{i=k+1}^m |g_i|}{(1 + \epsilon) \sum_{i=1}^m g_i^2 - \sum_{i=1}^k g_i^2} \right) \tag{6}$$

$$\sum_{i=1}^m p_i \le k(1 + \epsilon) \tag{7}$$

As such, the variance of the gradient in ID is bounded by Equation 7.

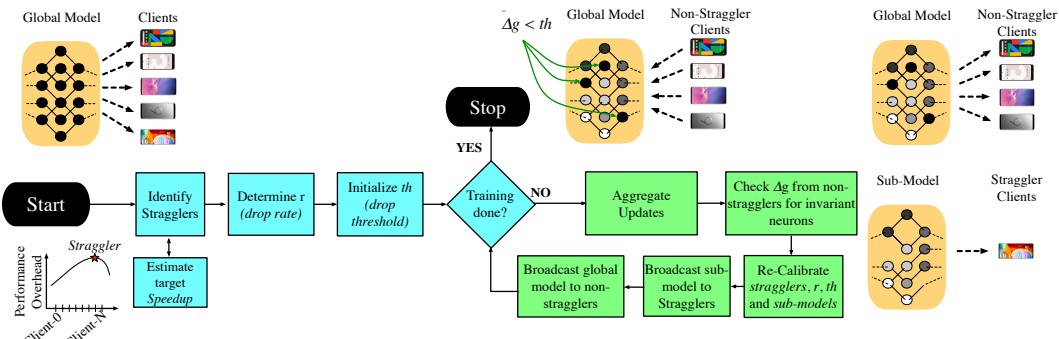

Figure 3: The workflow of FLuID. The non-stragglers are used to determine the neurons that are not updated within a set threshold. Thereafter, sub-models are dynamically created by dropping invariant neurons. These sub-models are sent to the straggler devices.

## 5 FLuID Framework

Figure 3 shows the workflow of Federated Learning using Invariant Dropout (FLuID). In FLuID, each calibration step includes straggler determination $(T)$, discovering invariant neurons$(IN)$ and drop threshold $(th)$, and sub-model extraction$(s_i)$. Currently, the calibration occurs per epoch, i.e., one training run over the complete client dataset. However, the frequency of calibration can be reduced if the invariant neurons and stragglers do not significantly change over steps.

Algorithm 1 outlines the FLuID framework. At the onset of training, FLuID identifies the stragglers. To achieve this, FLuID runs the global model on all clients, including stragglers, and measures the performance delay between the end-to-end training time of the slowest client ($T_{straggler}$) and the target time ($T_{target}$).End-to-end training includes upload/download latency and communication time.

$T_{target}$ is the desired training time FLuID aims to achieve for stragglers. FLuID assigns $T_{target}$ as the next slowest client's training time. This choice optimizes non-straggler idle time reduction. Note, FLuID can support any $T_{target}$ value. Setting $T_{target}$ lower than the next-slowest client's time offers no gain as non-stragglers cannot accelerate. Conversely, setting $T_{target}$ above the next-slowest client's training time leads to longer idleness and suboptimal performance. The required speedup for stragglers is calculated as $Speedup = \frac{T_{straggler}}{T_{target}}$. The initial calibration is carried out in lines 6-9 and 18-22 of the algorithm. Note, it takes a few epochs to calibrate the initial threshold, stragglers, and submodel. In subsequent calibration steps, the global server continues to measure the training time of clients and thereby identifies if there is any change in the straggler cohort.

**Tuning the performance of stragglers.** The sub-model size is determined by calculating the dropout rate $r$ as detailed in lines 18-21 of Algorithm 1. The value of $r$ is selected to ensure that the updated straggler training time, denoted as $T_{stragglernew}$, is close to the target training time ($T_{target}$). Figure 7 in Appendix A.3 demonstrates that across all evaluated datasets, the training time of all five mobile clients decreases linearly as the sub-model size decreases, and falls within 10% of the sub-model size. Using this insight, FLuID chooses an $r$ that is closest to the inverse of the speedup.

**Determining the drop threshold at runtime.** Once FLuID has identified the stragglers and the dropout rate, it iteratively adjusts the threshold to drop as many invariant neurons as possible. The design of FLuID is inspired by the preliminary results regarding the characteristics of invariant neurons and their impact on the model accuracy. In Appendix A.2, we present results to quantify the impact of the threshold value on the number of invariant neurons during training. We observe that, in order to obtain the desired accuracy, it is critical to select a threshold that yields a number of invariant neurons as close as possible to the number of neurons to be dropped for the sub-model.

Lets assume $w_{ijc}(t)$ represents the set of weights of the $i$th neuron in the $j$th layer for client $c$ after the training epoch $t$. The percent difference $g$ of the neuron, for each client $c$, is the minimum value of $g$ as denoted by: $g \geq \frac{w_{ijc}(t)-w_{ij}(t-1)}{w_{ij}(t-1)}$. Neurons that are potential candidates for dropping are those whose weight updates are within the threshold ($th$) compared to the previous calibration point. Unfortunately, determining the appropriate $th$ poses a few challenges.

---
**Algorithm 1** FLuID executing on Centralized Server
---

1: **Input**: $M(w)$      *# Model. $M(w_t)$ is model at step $t$*
2: **Output**: $S(w)$      *# Sub-models for stragglers*
3: $N \rightarrow$ All Clients, $T \rightarrow \emptyset$      *# T stragglers, N non-stragglers*
4: $IN \rightarrow \emptyset$      *# invariant neurons*
5: **while** not done **do**
6:      **if** $T$ is $\emptyset$ **then** {*# Straggler and dropout threshold intialization*}
7:          Broadcast $M(w_0)$ to each $C$
8:          Receive $M(w_1)$ from every $C$
9:          $th = \min \left( \frac{w_{ijct} - w_{ij(t-1)}}{w_{ij(t-1)}} \right)$ *# threshold for Invariant dropout*
10:      **else** {*# Straggler and dropout threshold recalibration*}
11:          $S(w_t) =$ sub_model_generation$(IN, M(w_t))$
12:          Broadcast $S(w_{ti})$ all $i \in T$
13:          Broadcast $M(w_t)$ all $i \wedge \notin T$
14:          Receive $\Delta(w_{t+1})$ from every $C$
15:          $M(w_{t+1}) =$ aggregate$(M(w_t), \Delta(w_{t+1}))$
16:          $IN =$ identify_invariant_neurons$(th, \Delta(w_{t+1}), N)$
17:      **end if**
18:      $T, N, L =$ determine_stragglers$(C)$ *# L = train latencies*
19:      $T_{target} =$ identify_next_slowest_client$(L)$
20:      $Speedup_i = \frac{T_{straggler_i}}{T_{target}}$
21:      $r_i, S_i =$ calculate_submodel_size$(Speedup_i)$ *# $r_i$ dropout rate*
22:      $th =$ increment_threshold$(r_i)$
23:      done = check_training_done$(M(w_t))$
24: **end while**

---

In order to identify neuron drop candidates, the global server cannot rely on updates from all clients since stragglers only train on and update the sub-model. Instead, the server takes advantage of the fact that non-stragglers train on the complete model and can identify neurons whose weight updates fall within the threshold $(th)$ for each calibration step. FLuID prioritizes dropping neurons on stragglers whose weight updates fall within the threshold $(th)$ for the majority of non-stragglers.

The initial threshold value $(th)$ in the FLuID framework is set as the average of the minimum percent update of all neurons in the initial few training epochs. The threshold is incrementally increased after each epoch until the number of neurons below the threshold is greater than or equal to the number of neurons to be left out of the sub-model. FLuID can have a different drop threshold for each layer. The algorithm targets neurons for dropping whose gradients consistently fall below the threshold over multiple epochs, prioritizing the elimination of non-critical neurons.

This entire process is repeated to recalibrate the stragglers, the drop rate $(r)$, and the threshold $(th)$.

## 6 Evaluation Setup

**Models and datasets.** We evaluate FLuID on three models and datasets as used by the prior works in the federated learning space [MGZP22, JWK$^+$22, HLA$^+$21, CKMT18].

The FEMNIST datasets consist of images of numbers and letters, partitioned based on the writer of the character in non-IID setting. We train a CNN with two 5x5 CONV layers with 16 and 64 channels respectively, each of them followed with $2 \times 2$ max-pooling. The model also includes a fully connected dense layer with 120 units and a softmax output layer. The model is trained using a batch size of 10 and a learning rate of 0.004.

The Shakespeare dataset partitions data based on roles in Shakespeares' plays in non-IID setting. We train a two-layer LSTM classifier containing 128 hidden units. We train the model with a batch size of 128 and a learning rate of 0.001.

The CIFAR10 dataset consists of images, partitioned using the same strategy as FjORD [HLA$^+$21] and the IID partition provided by the Flower [BTM$^+$20]. For the real-world mobile devices, we train on the VGG-9 model due to its ability to fit within the resource constraints of all the mobile phones we tested. VGG-9 [SZ15] model architecture consists 6 3x3 CONV layers (the first 2 have 32 channels,

Table 1: Software-Hardware specifications of clients

| Device | Year | Android Version | CPU (Cores) |
|---|---|---|---|
| LG Velvet 5G | 2020 | 10 | 1×2.4 GHz Kryo 475 Prime + 1×2.2 GHz Kryo 475 Gold + 6×1.8 GHz Kryo 475 Silver |
| Google Pixel 3 | 2018 | 9 | 4×2.5 GHz Kryo 385 Gold + 4×1.6 GHz Kryo 385 Silver |
| Samsung Galaxy S9 | 2018 | 10 | 4×2.8 GHz Kryo 385 Gold + 4×1.7 GHz Kryo 385 Silver |
| Samsung Galaxy S10 | 2019 | 11 | 2×2.73 GHz Mongoose M4 + 2×2.31 GHz Cortex-A75 + 4×1.95 GHz Cortex-A55 |
| Google Pixel 4 | 2019 | 12 | 1×2.84 GHz Kryo 485 + 3×2.42 GHz Kryo 485 + 4×1.78 GHz Kryo 485 |

Table 2: Accuracy comparison of Random Dropout, Ordered Dropout, and Invariant Dropout. The text in **bold** indicates instances when Invariant Dropout showcases the highest accuracy. ($\mu$ = mean, $\sigma$ = standard deviation, and $r$ = sub-model as a fraction of global model).

| Dataset | Dropout Method | $r = 0.95$ | | $r = 0.85$ | | $r = 0.75$ | | $r = 0.65$ | | $r = 0.5$ | |
|---|---|---|---|---|---|---|---|---|---|---|---|
| | | Accuracy ($\mu$) | $\sigma$ | Accuracy ($\mu$) | $\sigma$ | Accuracy ($\mu$) | $\sigma$ | Accuracy ($\mu$) | $\sigma$ | Accuracy ($\mu$) | $\sigma$ |
| Shakespeare | Random | 43.3 | 0.1 | 42.5 | 0.1 | 42.4 | 0.1 | 41.8 | 0.2 | 41.3 | 0.1 |
| | Ordered | 42.9 | 0.1 | 42.3 | 0.1 | 42.2 | 0.2 | 41.9 | 0.1 | 41.4 | 0.1 |
| | **Invariant** | **43.6** | 0.1 | **42.5** | 0.1 | **42.6** | 0.2 | **42.2** | 0.2 | **41.7** | 0.1 |
| CIFAR10 | Random | 57.5 | 0.1 | 56.8 | 0.2 | 57.0 | 0.2 | 57.2 | 0.2 | 57.2 | 0.3 |
| (VGG-9) | Ordered | 57.7 | 0.1 | 57.0 | 0.2 | 57.6 | 0.2 | 57.3 | 0.1 | 57.1 | 0.1 |
| | **Invariant** | **58.2** | 0.1 | **58.4** | 0.3 | 57.1 | 0.1 | **57.5** | 0.2 | **57.4** | 0.2 |
| FEMNIST | Random | 80.6 | 0.1 | 80.5 | 0.2 | 80.3 | 0.2 | 79.3 | 0.5 | 79.2 | 0.3 |
| | Ordered | 80.6 | 0.2 | 80.5 | 0.2 | 80.4 | 0.2 | 80.3 | 0.1 | 79.7 | 0.3 |
| | **Invariant** | **81.1** | 0.3 | **80.9** | 0.1 | **80.8** | 0.2 | **80.3** | 0.4 | **80.1** | 0.3 |

followed by two 64-channel layers, and lastly two 128-channel layers), two `FC` dense layers with 512 and 256 units and a final softmax output layer. We train the model with a batch size of 20 and a learning rate of 0.01. We conduct scalability experiments using the ResNet-18 model, and the results are presented in section 6.1. All the baseline dropout methods and Invariant Dropout are evaluated using the same setup.

**System Configuration.** Table 1 provides the details of the phones used for the experiments. We evaluate five clients and identify one straggler per training epoch. We connect all our client devices and our server over the same network. All the clients run on Android mobile phones from the years 2018 to 2020.

FLuID is implemented on top of the Flower (v0.18.0) [BTM+20] framework and TensorFlow Lite [Goo] from TensorFlow v2.8.0 [ABC+16]. Models are defined using TensorFlow's Sequential API, and then converted into `.tflite` formats.

**Evaluation metrics and baselines.** We compare the average training performance (wall-clock time) and accuracy for all workloads and experiments. In each evaluation round, clients receive the global model and report their evaluation accuracy and loss on local data to the server. The server calculates the distributed accuracy and loss by performing a weighted average based on the number of testing examples for each client. We compare FLuID with two established baselines: 1) Random Federated Dropout [CKMT18] and 2) Ordered Dropout from FjORD [HLA+21].

## 6.1 Results and Analysis

**Accuracy Evaluation** We compare accuracy of Invariant Dropout with two baselines, using different sub-model sizes. We trained the models for 100, 250, and 65 epochs for CIFAR10, FEMNIST, and Shakespeare datasets, respectively. Table 2 presents the average achieved accuracy ($\mu$) along with the standard deviation ($\sigma$) for the three datasets. Specifically, Invariant Dropout outperforms Random Dropout across all three datasets. Compared to Random Dropout, our work achieves a maximum accuracy gain of 1.6% points and on average 0.7% point higher accuracy for FEMNIST, 0.6% point higher accuracy for CIFAR10, and 0.3% point higher accuracy for Shakespeare datasets.

Invariant Dropout also achieves a higher accuracy against Ordered Dropout across all three datasets, with a maximum increase in accuracy of 1.4% and an average increase of 0.3% for FEMNIST, 0.4%

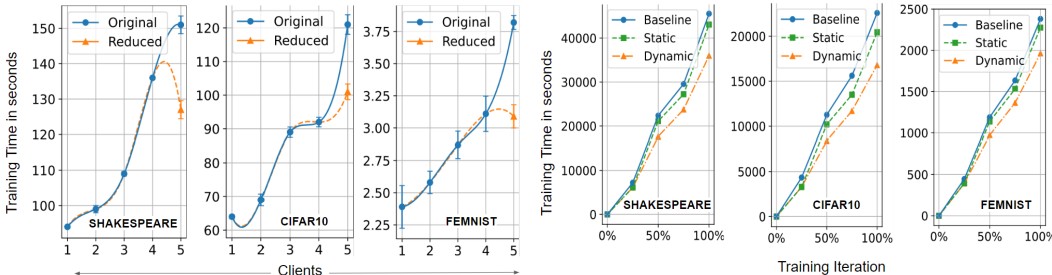

(a) Training time for stragglers *before and after* FLuID.    (b) Training time reduction with varying stragglers.

Figure 4: Performance evaluation of FLuID

for CIFAR10, and 0.4% for Shakespeare. The accuracy improvements of Invariant Dropout are statistically significant ($\alpha < 0.05$). Moreover, Invariant Dropout shows less variation in accuracy between runs of the same sub-model size and across all sub-model sizes. This is because invariant Dropout eliminates only invariant neurons, which have little impact on the final model efficiency.

**Computational Performance Evaluation** Figure 4a shows FLuID's capability to effectively select the sub-model, resulting in a significant reduction in the straggler's training time that almost matches the next slowest client. In the absence of FLuID, the straggler's training time is typically 10% to 32% longer than the target time. However, after applying FLuID, the straggler's training time is within 10% of the target time. Furthermore, it is observed that the overall accuracy of the global model tends to be higher when stragglers train with larger sub-models. Consequently, FLuID selects the largest possible sub-model size that minimizes training time variance across clients. Notably, the performance improvement, unlike accuracy, is influenced by the size of the sub-model and not the specific dropout technique employed. For a more detailed analysis of the impact of sub-model size on training time, please refer to Appendix A.3.

**Varying stragglers at runtime.** FLuID can recalibrate stragglers. During the experiments in Table 2, we observe that the performance of our mobile clients remained relatively stable, without significant changes in the straggler. However, to evaluate the impact of varying conditions at runtime, we randomly executed certain clients at different points in the training process (25%, 50%, and 75% marks). This was achieved by enabling a client to run the training program as a background process between the specified periods. We observed that FLuID successfully adapted to the variation in stragglers during runtime. Figure 4b demonstrates the overall training time for this experiment. On average, the FLuID framework achieved 18% to 26% faster training time compared to the baseline, and 14% to 18% faster training time compared to selecting a static straggler throughout the entire training process. All results include the overhead of FLuID, which, importantly, is not significant. We observe, FLuID calibration process takes significantly less time (less than 5%) compared to the actual training time. This is because the additional computations required for threshold and sub-model calibration are performed centrally on the server, rather than being distributed to edge devices.

**Scalability study.** To assess the scalability of FLuID, we conducted experiments using simulated clients ranging from 50 to 100. Each machine runs 10 to 20 clients in parallel. Among all clients, we identified the slowest 20% as stragglers. We further extend the experiment to run CIFAR10 with ResNet-18 as these are emulated clients on a server and can support relatively larger models than mobile devices. Figure 5 shows the accuracy performance across all three datasets. Overall, Invariant Dropout consistently outperforms Ordered and Random Dropout and maintains a better accuracy profile similar to Table 2. In addition, our dropout technique performs significantly better than completely excluding stragglers from the training process.

We further extended the experiment to assess the scalability of FLuID by 1) clustering stragglers into multiple sub-model sizes (Appendix A.4), 2) Exploring the impact of varying ratios of stragglers in the system (Appendix A.5), and 3) a scalability study with 1000 clients where client sampling is employed as we cannot check for stragglers across 1000 devices individually (Appendix A.6). We demonstrate that FLuID can scale to scenarios involving multiple stragglers with varying computational capabilities by tailoring sub-model sizes for each client. Moreover, when compared to state-of-the-art dropout techniques, Invariant Dropout achieves high accuracies, even as the percentage of stragglers in the system scales up.

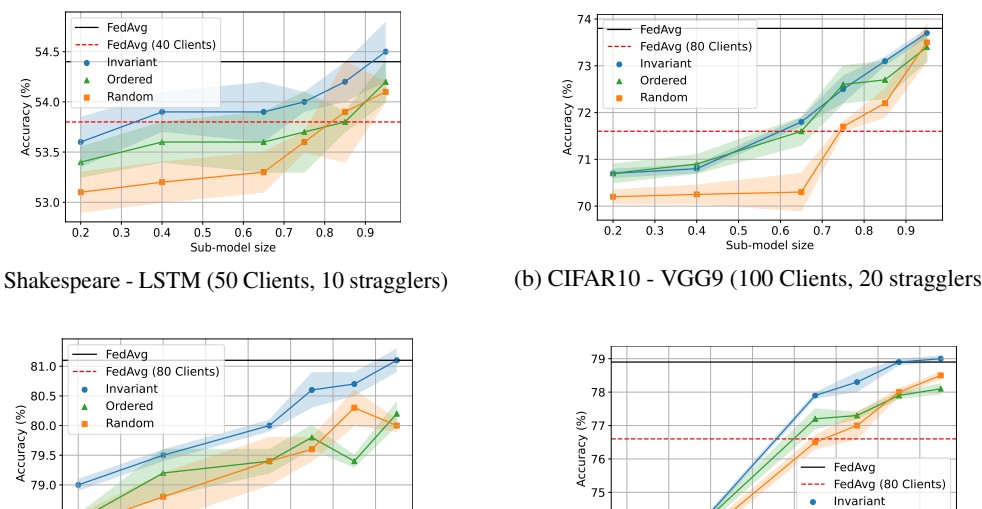

(a) Shakespeare - LSTM (50 Clients, 10 stragglers)

(b) CIFAR10 - VGG9 (100 Clients, 20 stragglers)

(c) CIFAR10 - ResNet18 (100 Clients, 20 stragglers)

(d) FEMNIST - CNN (100 Clients, 20 stragglers)

Figure 5: The accuracy comparison of Invariant Dropout with Ordered and Random Dropout as we scale to 50-100 clients with 20% of the slowest clients being stragglers.

# 7 Limitations and Future Work

Although FLuID is able to mitigate some impact of the stragglers, it does incur minimal overhead to handle stragglers and maintain system performance. Our evaluation takes this into account but this overhead may increase if straggler performance constantly changes.

FLuID currently only uses pre-defined sub-model sizes mapped to straggler performance, which keeps the framework lightweight and avoids high overhead. However, for future works with varied edge devices, fine-grained sub-model determination may further enhance the work.

# 8 Conclusions

Due to rapid technological advancements and device variability in handheld devices, system heterogeneity is prevalent in federated learning. Straggler devices, which exhibit low computational performance, act as the bottleneck. In this paper, we address these issues by introducing a novel dropout technique called Invariant Dropout. Invariant Dropout dynamically creates customized sub-models that include only the neurons exhibiting significant changes above a certain threshold. We build a framework, FLuID, which adapts to changes in stragglers as runtime conditions shift. FLuID effectively mitigates the performance overheads caused by stragglers while also achieving a higher accuracy compared to state-of-the-art techniques.

# 9 Acknowledgements

We thank the anonymous reviewers for their insightful comments. This research was supported in part through computational resources provided by Advanced Research Computing at the University of British Columbia (UBC) [soc]. This work was partially supported by Gift from Google and the Natural Sciences and Engineering Research Council of Canada (NSERC) [funding reference number RGPIN-2019-05059]. The work, in part, was supported by Georgia Tech School of Electrical and Computer Engineering and School of Computer Science. The views and conclusions contained herein are those of the authors. They should not be interpreted as necessarily representing the official policies or endorsements, either expressed or implied, of Georgia Tech, Microsoft, and UBC.

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

# A Appendix

## A.1 Evolution of Invariant Neurons

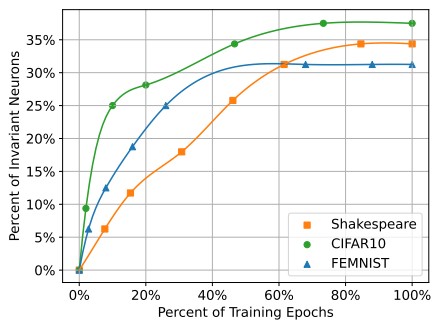

Figure 6: The percentage of 'invariant' neurons as the number of training rounds vary

In this section, we provide an example that some neurons in the server are trained quickly and vary only below a threshold in later iterations. Figure 6 shows the percentages of invariant neurons as the number of training epochs increases. Even after only 30% of the training rounds are completed, 15%-30% of the neurons become invariant across CIFAR10, FEMNIST, and Shakespeare datasets For this example, we choose thresholds of 180%, 10%, and 500%, respectively, for these three datasets and compute their invariant neurons. Sending invariant neurons over to the straggler provides no utility; therefore, these neurons can be dropped. Our work FLuID builds upon this insight.

## A.2 Choosing Suitable Threshold

Each model has different characteristics in terms of the magnitude of neuron updates. Therefore, choosing different threshold values results in a different number of neurons classified as invariant. We expanded on our initial findings and studied the effect of threshold value on the number of invariant neurons during training. As expected, a higher threshold value leads to a higher percentage of invariant neurons. Table 3 presents the percentage of invariant neurons observed at different threshold values, and the overall training accuracy of the FEMNIST model, using a sub-model size of 0.75 for the stragglers.

Table 3: Threshold vs accuracy results

| Threshold value (%) | Percentage of Invariant Neurons (%) | Accuracy (%) |
|:---:|:---:|:---:|
| 1 | 3 | 80.1 |
| 3 | 6 | 80.3 |
| 5 | 13 | 80.5 |
| 7 | 18 | 80.7 |
| 8 | 22 | 80.7 |
| 10 | 31 | 80.5 |

We observe that to obtain the desired accuracy and mitigate performance bottlenecks of stragglers, it is critical to choose the threshold that has the closest number of invariant neurons as the number of neurons to be dropped for the sub-model. The FLuID framework can automatically tune the threshold for the desired model based on the straggler performance, as described in Section 5.

## A.3 Impact of Sub-Model Size on Training Time

In this section, we present evidence that there is a linear relationship between client training time and sub-model size. We evaluated the training time of 5 Android-based mobile phones from 2018 to 2020 outlined in Table 1. The training time is expressed in the percentage of the training time for the

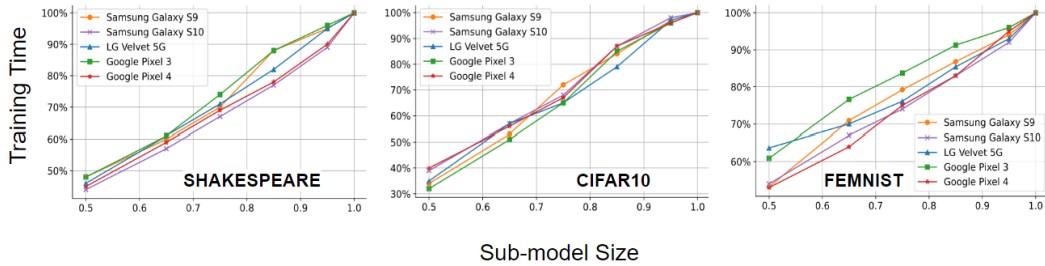

Figure 7: Linear relationship between training time and model size across all three datasets

full model size ($r = 1.0$). Across CIFAR10, FEMNIST, and Shakespeare, the training time of all five mobile clients decreases linearly as the sub-model size decreases and falls within 10% of the sub-model size. Using this insight, FLuID selects a sub-model size $r$ as the available sub-model, the size that's closest to the inverse of $Speedup$.

## A.4 Additional Experiments with Scalability Studies.

In this section, we show that in the case that the network has multiple stragglers, FLuID does not assume that all stragglers have similar capabilities or select a sub-model for all stragglers based on the slowest device. FLuID can select sub-model sizes for each straggler client based on each client's own capabilities. In this experiment, we cluster devices of similar capabilities into four groups of sub-model sizes. Table 4 presents the accuracy when stragglers are assigned to 4 equal-sized clusters (sub-model size 0.65, 0.75, 0.85, 0.95). The overall accuracy generally lies between assigning sub-model sizes of 0.75 and 0.85 for all stragglers. This way, FLuID can achieve a higher training accuracy with a shorter training time, even with stragglers that are initially more than 35% slower than non-straggler devices.

Table 4: Accuracy comparison of Random Dropout, Ordered Dropout, and Invariant Dropout as we cluster stragglers into different sub-model size groups.

|  | Random | Ordered | Invariant |
|---|---|---|---|
| CIFAR10 | 71.7 | 72.3 | 72.7 |
| FEMNIST | 77.5 | 77.4 | 78.2 |
| Shakespeare | 53.8 | 53.9 | 54.1 |

## A.5 Additional Experiments with Varying Straggler Percentages

In this section, we show that FLuID is capable of handling multiple ratios of stragglers. We have run additional experiments to explore the impact of different ratios of stragglers. One common trend we observed across state-of-the-art techniques and FLuID is that accuracy decreases as the ratio of stragglers is increased as more of the devices are now being trained only on the sub-model. Nonetheless, in all the cases, Invariant Dropout offers the highest accuracy because it is aware of the neuron gradient changes and only drops the least changing neurons. The accuracy results of varying the straggler ratios while using 0.75-sized sub-models are summarized in Figure 8.

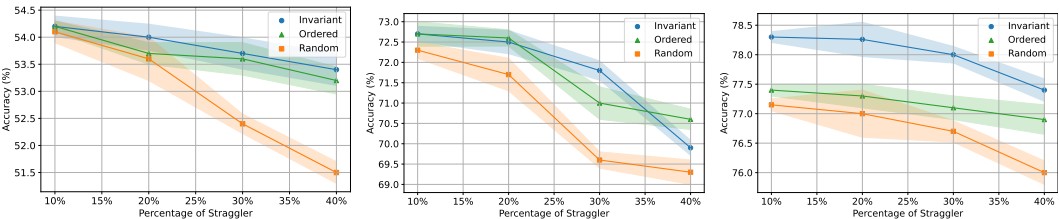

(a) Shakespeare - LSTM (50 Clients)  (b) CIFAR10 - VGG9 (100 Clients)  (c) FEMNIST - CNN (100 Clients)

Figure 8: Accuracy of varying the straggler ratios from 10% to 40% with 0.75 sub-model size

### A.6 Scalability of FLuID with Client Sampling

As the federated learning system scales, FL servers sample a subset of clients participating in each training round. At any point in training, FLuID is capable of recalibrating stragglers and supports dynamic changes during runtime. The ability to adjust to a different set of clients and identify stragglers in every training round enables FLuID to easily incorporate client sampling into its process. We scaled FLuID to 1,000 clients with the FEMNIST dataset for 500 global training rounds. We run with a client sampling ratio of 10%, as used by the prior works in federated learning spaces such as FjORD [HLA+21]. We present the accuracy results in Table 5 against each sub-model size for Invariant Dropout and the baseline techniques. Invariant Dropout maintains a better accuracy profile than the baselines even when scaled up to 1000 clients while incorporating client sampling.

Table 5: Accuracy comparison of Random Dropout, Ordered Dropout, and Invariant Dropout as for FEMNIST with 1000 clients and client sampling of 10%.

|           | r=0.95 | r=0.85 | r=0.75 | r=0.65 | r=0.40 |
|-----------|--------|--------|--------|--------|--------|
| Random    | 87.9   | 87.5   | 87.5   | 86.9   | 85.7   |
| Ordered   | 87.8   | 88.0   | 87.5   | 87.3   | 87.0   |
| Invariant | 88.1   | 88.2   | 88.0   | 87.7   | 87.2   |

