# OpenReview forum: "FLuID: Mitigating Stragglers in Federated Learning using Invariant Dropout"
_NeurIPS.cc/2023/Conference — NeurIPS 2023 poster_

### Official Review · Reviewer_DQm5 · 2023-07-06

**Soundness:** 2 fair
**Presentation:** 3 good
**Contribution:** 2 fair
**Rating:** 5
**Confidence:** 4

**Summary:**

The paper proposes (Federated Learning using Invariant Dropout) FLuiD to address the straggler problem in FL. Due to the presence of system heterogeneity, straggling nodes (nodes with resource constraints) become a bottleneck in the FL training process. In this paper, the authors propose a FLuID to address this problem. First, the identification of invariant neurons, neurons that quickly optimize and remain relatively stable for the remainder of the training. Second, the straggling devices participating in the training. This information will be used to “prune” the global model and send a subset model to the straggling devices in order to efficiently utilize the available computation and communication resources.



**Strengths:**

+ The straggler experiments included setups conducted using mobile devices.
+ The experiments show the proposed invariant dropout speeding up execution time relative to ordered dropout and random dropout.


**Weaknesses:**

- It appears most of the experiments involved only 5 devices. While it is commendable that the experiments involved hardware, it is not clear to claim that the accuracy margins would hold as the number of clients increases, as is expected in practical FL environments. The scaled up experiments involving 50 to 100 clients can easily be scaled up to 1000 - 3000 clients in FEMNIST for instance. This might require an FL simulation environment, but it would give a good sense of how the proposed scheme yields improved training time and accuracy.

- FL usually requires client sampling in each round. This is necessary as the system scales, and not all clients can participate in each round. I didn’t see the client sample ratio discussed. Thus I assume, in your experiments, all clients are participating in each training round. If that’s the case, how will client sampling in each round affect the dropout scheme? It is not clear if the invariant dropout scheme will maintain its performance margin if client sampling is incorporated into the training process.


**Questions:**

- How the proposed approach is scalable with increasing the number of clients (clients can easily be scaled up to 1000 - 3000 clients in FEMNIST for instance.)?
- Does the invariant dropout scheme will maintain its performance if client sampling is incorporated into the training? Clarification on this and further experiments would be helpful.
- The authors should enhance the plot labels in Fig. 5

**Limitations:**

- Scalability is a major limitation that is not discussed and experimented well.
- Sampling during the training process needs to be addressed clearly.

---

> ### Author Rebuttal · Authors · 2023-08-09
>
> We thank the reviewer for the insightful comments, which have enabled us to add scalability analysis and sampling to the paper.
>
> **Question 1: Scalability**
>
> We scale FLuID to **1000 clients** with the FEMNIST dataset for 500 global training rounds. We run with a **client sampling ratio of 10%**, as used by the prior works in federated learning space such as FjORD. We run these experiments on a private cloud cluster of Intel Xeon Silver CPUs and NVIDIA Tesla V100 GPUs. We emulate 20 clients on each machine using the Flower framework, as detailed in the scalability studies in Section 6.1.
> We remain consistent with our other evaluations in Section 6.1, identifying the slowest 20% of clients as stragglers. At each global round, we randomly sample 10% of the available clients for the training round. FLuID maintains a record of the stragglers' cohort among all available clients. FLuID monitors the training times of the clients within the sampled group after every global training round. It updates the straggler record if any changes or new stragglers are identified.
>
> Below, we present the accuracy results against each sub-model size for Invariant Dropout and the baseline techniques. Invariant dropout maintains a better accuracy profile than the baselines even when scaled up to 1000 clients while incorporating client sampling. We will add these experimental results in the final paper to showcase the efficacy of FLuID as it scales to a large number of devices for the evaluated datasets.
>
> Dropout Method | r=0.95| r=0.85| r=0.75| r=0.65| r=0.40|
> ---------------|-------|--------|-------|------|-------|
> Random|87.9|87.5|87.5|86.9|85.7
> Ordered|87.8|88.0|87.5|87.3|87.0
> Invariant|88.1|88.2|88.0|87.7|87.2
>
> **Question 2: Client sampling**
>
> At any point in training, FLuID is capable of recalibrating stragglers and supports dynamic changes during runtime. This allows FLuID to easily incorporate sampling in its process, as showcased by the new results presented here. If certain clients become unavailable or device properties change at any moment, FLuID can take that into account, as discussed in Section 6.1 and Figure 4b.
>
> However, all clients participated in each training round for the experiments included in the paper.
> We scale FLuID to execute on 1000 clients in a system that emulates each client on the server described in Section 6 of the paper. We perform the experiments with the FEMNIST dataset with a client sampling ratio of 10%, as used by the prior works in federated learning space such as FjORD. We will add these experimental results **as presented above** in the final paper to showcase the efficacy of FLuID to incorporate client sampling as the number of devices grows.
>
> **Question 3: Enhancing Fig. 5**
>
> We will enhance all the plot labels to ensure it is legible.

---

### Official Review · Reviewer_a2Bu · 2023-07-07

**Soundness:** 3 good
**Presentation:** 2 fair
**Contribution:** 2 fair
**Rating:** 5
**Confidence:** 3

**Summary:**

The paper focuses on the issue of mitigating stragglers in a heterogenous FL environment through dynamic load balancing, introducing a technique called Invariant Dropout and an adaptive training framework called FLuID. Invariant Dropout dynamically creates customized sub models that include only the neurons exhibiting significant changes over a certain threshold. Through experimental evaluation, the papers shows that this approach mitigates performance overheads caused by stragglers while also achieving higher accuracy compared to the state-of-the-art technique - Ordered Dropout.

**Strengths:**

- Up to 18% speedup in performance
- Up to 1.4 pp improvement in accuracy over state-of-the-art Ordered Dropout
- Evaluation is detailed - on 3 models and datasets and compared with two techniques: Random and Ordered
- Improves on previous work which have drawback of incurring training bias, creating performance centric sub models, or entirely reconstructing the sub model


**Weaknesses:**

- Accuracy gains compared to Ordered and Random across datasets and models, while statistically significant may not justify the additional complexity of this system.
- There is scope to improve the presentation
   - In the lines 40-47 2 sentences are repeated
   - Figure 2b has no label on the x axis


**Questions:**

N/A

**Limitations:**

In addition to the weaknesses mentioned above, the authors note two limitations
- Overhead to handle stragglers and maintain system performance which may increase with changes in straggler performance
- Currently the system only uses pre-defined sub model sizes mapped to straggler performance which keeps the framework lightweight, but for varied edge devices, fine grained sub model determination will be required.

The impact of this fine grained approach on the overhead needs to be measured to check for its suitability on different edge devices. If the overhead increases, the performance improvements noted here may diminish.

---

> ### Author Rebuttal · Authors · 2023-08-09
>
> We thank the reviewer for their insightful comments. We answer the most pressing questions first, followed by the remaining questions.
>
> **Weakness 1: Accuracy Gains of FLuID**
>
> We have implemented FLuID as a lightweight system with minimal overheads. We empirically observe that the FLuID calibration process takes significantly less time (less than 5%) than the actual training time. Our evaluation takes these overheads into account while reporting the performance benefits. As we observe more heterogeneity in the system of devices, frameworks like FLuID will play a crucial role in mitigating performance bottlenecks.
>
> **Limitations**
>
> FLuID can scale to support more submodel sizes without significantly increasing overhead. As discussed in section 5, line 205, the sub-model size selection process is straightforward.  FLuID simply chooses a sub-model size closest to the inverse of the required speedup. Therefore, even with the addition of more sub-model sizes, the processing overhead complexity does not experience a significant increase. Notably, on real mobile phones, we have observed that the FLuID calibration process takes less than 5% of the training time.
>
> **Weakness 2: Improving the presentation**
>
> We will clean the text to remove the repetition and ensure all figures have labels.

---

### Official Review · Reviewer_yFRv · 2023-07-08

**Soundness:** 3 good
**Presentation:** 4 excellent
**Contribution:** 3 good
**Rating:** 6
**Confidence:** 4

**Summary:**

FLuID authors tackle a straggler problem in Federated Learning, where a central model is trained across a set of heterogeneous devices.
This problem is particularly challenging when performance capabilities at training time are actually variable. This variable heterogeneity at training time requires a mechanism that can change the model sent to a client per round, in order to create a control mechanism that mitigates the clients’ straggler effect. FLuID achieves this with a proposed Invariant Dropout method.
The Invariant Dropout technique quite simply refers to identifying neurons the updates for which have become minimal (close to invariant) over some number of rounds. This observation is used to decide to exclude such neurons (or “drop” them) from training for a given client. Neurons can be ranked based on the extent to which they are “invariant”, and a line can be drawn as a function of computation budget to determine how many neurons to drop.

FLuID builds on several technical challenges:
1. identifying invariant neurons
2. identifying stragglers
3. determining a subset of the model weights to send to each client/straggler

FLuID is able to match or exceed accuracy across several datasets. Performance includes real-world mobile client evaluation. Straggler effects are mitigated using Invariant Dropout.

**Strengths:**

+ FLUid tackles a quintessential problem of stragglers in FL, which very practical and pervasive
+ FLUid avoids using asynchronous aggregation techniques, which jeopardize model convergence. It sticks with synchronous aggregation
+ the dropout technique is well known in DNN training and has been shown to improve performance, when random dropout is used
+ The dynamic nature of invariant dropout is effective and can adapt to dynamically changing client compute (or communication) capacity. Support for this dynamic heterogeneity is rare in literature
+ the paper is well written
+ the experimental methodology is well executed

**Weaknesses:**

* the idea is very simple, some could consider the adaptation of dropout from previous literature to FL setting incremental? I’m certainly familiar with this idea and surprised this hasn’t been published yet.
* this paper compares to different types of dropout as baselines, but doesn’t consider other published baselines, such as SuperFed [1]:
    * https://arxiv.org/abs/2301.10879

[1] SuperFed, https://arxiv.org/abs/2301.10879, 26 Jan 2023

**Questions:**

Questions:
* It would be interesting to get your perspective on comparison between FLuID and SuperFed [1]. SuperFed proposes sending clients subgraphs of the central model, which is isomorphic to structured dropout. SuperFed does NOT track neuron invariance, of course, but the idea of using structured dropout to send a smaller model to the client is there. In its conclusion, SuperFed mentions “adjusting .. for client resource constraints (e.g., compute, bandwidth, memory capacity)”.

[1] SuperFed, https://arxiv.org/abs/2301.10879, 26 Jan 2023

**Limitations:**

yes, credit to authors for pointing out limitations: minimal overhead to handle stragglers and degradation of system perf in the worst case when the dynamics of client capacity variability causes instability.

---

> ### Author Rebuttal · Authors · 2023-08-09
>
> We thank the reviewer for their comments. We provide the comparison against SuperFed below:
>
> Both SuperFed and FLUID propose sending a subset of a global model to edge clients, and both frameworks can send submodels of varying sizes to each client. However, these two frameworks differ significantly in two main aspects: their objectives and the approach used to form and distribute the subnets to clients.
>
> Firstly, the objective of SuperFed is to co-train a global "family of models" in a federated fashion, which ultimately reduces the cost of training multiple global models. On the other hand, FLuID aims to mitigate the performance bottleneck caused by stragglers in federated learning while training for a single global model. As a result, these differences in objectives lead to distinct design approaches. **In SuperFed, all clients receive subnetworks of different sizes, while in FLuID, most clients train on the full global network, and only a smaller percentage of clients (stragglers) train with sub-models.**
>
> Secondly, the formation and distribution of sub-models also exhibit significant differences between the two frameworks. SuperFed focuses on optimizing the spatial and temporal distribution of subnetworks, ensuring that each subnetwork receives equal exposure to all data partitions. Additionally, it keeps track of the number of times each client has been assigned to the smallest and largest subnetworks. During each round, SuperFed utilizes this information to assign the smallest, largest, and random subnetworks to the clients. **In contrast, Fluid takes a different approach, as pointed out by the reviewer, by tracking neuron invariance to generate submodels of sizes specifically optimized for the computational capabilities of the straggler device.**
>
> We find SuperFed to be highly interesting, focusing on co-training model families in a cost-efficient federated fashion. The insights obtained from FLuID could prove valuable for the future advancement of SuperFed. By considering system heterogeneity and individual device capabilities in the load balancing and subnetwork distribution techniques, these frameworks can achieve improved performance and significant savings in training costs.

---

### Official Review · Reviewer_sKfw · 2023-07-23

**Soundness:** 2 fair
**Presentation:** 2 fair
**Contribution:** 1 poor
**Rating:** 5
**Confidence:** 3

**Summary:**

The paper presents a framework, FLuID, for cross-device federated learning, where some of the clients are “stragglers”. The training time of these stragglers is significantly higher, hence they dictate the overall training time.

FLuID uses Invariant Dropout to dynamically reduce the stragglers’ training time, hence alleviating their overall FL training time degradation.


**Strengths:**

The authors tackle a very important problem in cross-device federated learning. Clients are usually edge and mobile devices that are running several processes and applications at the same time. Therefore, the training time varies as compared to dedicated servers used in cross-silo FL setups.

Therefore, proposing a dynamic approach to identify the stragglers and the appropriate dropout rate is of major importance in this domain.


**Weaknesses:**

While this paper targets a very important and relevant problem in the FL domain, I have concerns regarding the proposed framework, as detailed below:

* How FLuID measures the training time of the stragglers, and how can it distinguish between stragglers due to high training time, and high upload/download latency? FLuID targets the former, while the latter requires a different solution than proposed in this paper.

* Some of the design decisions/details lack further explanations. For instance, the setting of T_target and th (see more about it in the Questions section).

* The number of invariants is bounded (at any given training round for any given model), while “Speedup” (as defined in the paper in line 197) is generally unbounded (depending on the stragglers’ training times). Therefore, FLuID might result in an aggressive dropout rate, and thus diminishing return.

* The evaluation is insufficient:

    (1) It includes small and relatively “easy” datasets to learn, as well as small ML models.

    (2) The obtained accuracy improvement is quite negligible.

    (3) Some of the evaluation setup information is missing. Mostly the distribution of the stragglers' training time, see also question (1) in the Questions section.


**Questions:**

(1) Lines 196-198: “T_target is set equal to the training time of the ‘next slowest client’ … Thus speedup ensures optimal utilization of available clients”, can you explain this argument? How does such a setting of T_target ensure optimal utilization of the clients, regardless of the stragglers’ training time distribution?

(2) Line 200: how does the server measure the training time of clients? The server is aware of the end-to-end latency of given clients that generally depends on the client’s (1) downlink bandwidth and latency, (2) training time, and (3) uplink bandwidth and latency.

(3) Line 11 in Algorithm 1 is unclear: what’s “inv”?

(4) The discussion about setting th in lines 209-219 is unclear, is it a heuristic? Does it have any theoretical properties/guarantees?

---

> ### Author Rebuttal · Authors · 2023-08-09
>
> We thank the reviewer for their insightful comments. We answer the questions in order.
>
> **Weakness 1: Training time of stragglers**
>
> FluID considers the end-to-end latency, upload/download latency, communication time, and training time of the device to determine if it is a straggler.
>
> However, similar to prior works in the area, such as Federated Dropout (Caldas et al., 2018) and FjORD(Horvath et al., 2021), the FLuID infrastructure is built to reduce the compute and communication load by only training on sub-models. FLuID focuses on identifying stragglers based on hardware processing capabilities. This approach effectively decreases the computation load on the device, focusing on training a sub-model and reducing the communication load by synchronizing fewer parameters.
>
> We empirically observe that download/upload latencies are similar and **do not pose significant bottlenecks** for any device or dataset. For both CIFAR10 and Shakespeare datasets, the download/upload time accounts for less than 3% of the end-to-end training latency. Similarly, in the case of FEMNIST, even with relatively short actual model training times, the download/upload time remains within 15% of the end-to-end latency.
>
> FLuID assigns T_target as the "next slowest client's training time." This choice optimizes non-straggler idle time reduction. FLuID is flexible to accommodate various t_target values. However, setting T_target lower than the "next-slowest client's" training time offers no gain as non-stragglers cannot accelerate. Conversely, setting t_target above training time leads to longer idleness and suboptimal performance.
>
> **Weakness 3: Unbounded Speedup**
>
> FLuID enforces a minimum model size to ensure accuracy is not significantly impacted. Consequently, in scenarios where the straggler's performance is exceptionally slow, FLuID might not **completely** eliminate the performance bottleneck caused by the straggler. Nevertheless, even in cases with high skewness in performance, FLuID will still alleviate some performance bottlenecks and reduce idle time for non-stragglers. Throughout our evaluations, we maintain a lower bound for the model size at 20%.
>
> **Weakness 4-(1) Evaluation datasets**
>
> We evaluated FLuID on established models, datasets, and settings as used by the prior works in federated learning space such as  Federated Dropout (Caldas et al., 2018), FjORD(Horvath et al., 2021), Adaptive Gradient Sparsification for Efficient Federated Learning (Han et al., 2020), PruneFL (Jiang et al.,2022), and FLANC(Mei et al., 2022).
>
> **Weakness 4-(2) Impact of FLuID**
>
> The primary goal of FLuID is to address the performance bottlenecks caused by stragglers in a federated learning scenario without compromising the model's accuracy. Compared to the state-of-the-art federated dropout techniques, FLuID offers up to 18% speedup in training time. FLuID introduces a novel dropout technique that considers neuron updates and performs dropout on invariant neurons to achieve this. This approach, moreover, results in accuracy improvements of up to 1.4%.
>
> **Weakness 4-(3) Distribution of the stragglers' training time**
>
> We apologize for the confusion on the evaluation setup. Figure 2a shows the per epoch training time distribution across the mobile devices in the log scale. The standard deviation between the training times of each client is 0.5, 22, and 21 seconds for FEMNIST, CIFAR10, and Shakespeare, respectively.  We will add these details to the paper.
>
> **Question 1: Setting of T_target**
>
> We will include the following explanation in the paper.
> FLuID aims to determine the optimal training time for straggler devices to mitigate non-straggler idle time in federated learning efficiently. To achieve this, we set a maximum training window while minimizing model dropout that effectively manages straggler impact without sacrificing accuracy. Notably, FLuID can enhance accuracy by up to 1.4% points compared to leading federated dropout techniques.
>
> **Question 2: Training time measurements**
>
> The **reported training time** is each client's **actual wall clock training time**. The server's end-to-end training latency is also measured by taking the time difference between when it sends the global model and receiving training results from each client. Thus, this training time includes the upload/download latency, training time, and communication time. We will clarify these details in the paper.
>
>
> **Question 3: Line 11 in Algorithm 1 “inv”**
>
> The inv is a typo. Inv instead is the variable IN in line 4 and line 16 of the algorithm. We will fix this in the final version of the paper.
>
> **Question 4:  Threshold selection**
>
> The threshold selection is guided by a heuristic approach rooted in preliminary results and backed by evaluation data. We will clarify this in the paper.
>
> The design of FLuID is inspired by the preliminary results regarding the characteristics of invariant neurons and their impact on the model accuracy. We have presented such preliminary results in Appendix A.1, Figure 6. These results show the number of invariant neurons across the training process. After only 30% of the training rounds are completed, 15%-30% of the neurons exhibit invariance across CIFAR10, FEMNIST, and Shakespeare datasets.
>
> A neuron is invariant if its update is below the threshold value. Different models exhibit distinct characteristics concerning the magnitude of neuron updates. Therefore, choosing different threshold values will lead to different numbers of neurons being categorized as invariant. In Appendix A.2, we showcase the threshold sweep and observe its impact on invariant neurons observed in the training process. Intuitively, the percentage of invariant neurons increases as the threshold value increases. Table 3 shows the percent of neurons observed with increasing threshold value as invariant and the overall training accuracy of the model trained for FEMNIST, with a sub-model size of 0.75 for the stragglers.

---

> > ### Comment · Reviewer_sKfw · 2023-08-13
> >
> > The authors have addressed most of my concerns.
> >
> > However, they aren't addressed in the submission, hence it's crucial to revise the paper in case of acceptance accordingly.
> >
> > Based on that assumption, I'm upgrading my ratings to 5.

---

> > > ### Author Response · Authors · 2023-08-13
> > >
> > > We thank the reviewer for their comments. We will incorporate all the clarifications in the final version of the paper.

---

### Decision · Program_Chairs · 2023-09-21

**Decision:**

Accept (poster)

**Comment:**

The reviewers generally agreed that the paper addresses an important problem and proposes a simple yet effective solution.